# Experimental Study on Gully Erosion Characteristics of Mountain Torrent Debris Flow in a Strong Earthquake Area

**Jiqin Zhang** [1], **Dengze Luo** [2], **Hongtao Li** [1,3], **Liang Pei** [1,3] and **Qiang Yao** [1,3,*]

1   College of Water Resource and Hydropower, Sichuan University, Chengdu 610065, China
2   Yalong River Hydropower Development Company, Ltd., Chengdu 610000, China
3   State Key Laboratory of Hydraulics and Mountain River Engineering, Sichuan University, Chengdu 610065, China
*   Correspondence: yaoqiang777@scu.edu.cn

**Abstract:** In recent years, as the frequency of debris flow outbreak in strong earthquake areas has increased and the scale has been expanding, in order to explore the erosion characteristics of debris flow, a lateral erosion flume model experimental device has been designed, and 18 groups of incomplete orthogonal experiments have been carried out, with a unit weight of debris flow of $1.6{\sim}2.0$ g/cm$^3$, a content of fine particles in the accumulation of $0{\sim}28.82\%$, and a longitudinal slope gradient of the gully of $8°{\sim}20°$ as variables. The results show that the erosion width, erosion depth, and erosion volume decrease with the increase in fluid bulk density and increase with the increase in gully slope. When the longitudinal slope of the gully was $16°$, the sediment with $11.40\%$ fine particles had the strongest erosion effect, indicating that more or less fine particles are not conducive to the occurrence of lateral erosion of the gully. Finally, through multi-factor variance analysis, it was found that the order of the three factors on the gully lateral erosion degree from strong to weak is: debris flow unit weight, gully slope, and accumulation grading. The analysis results further showed that the unit weight of debris flow has the greatest impact on the erosion degree of the side slope, which is consistent with the experimental results. The research results have important reference significance for revealing the mechanism of lateral erosion and improving the level of debris flow disaster prevention in strong earthquake areas.

**Keywords:** debris flow; lateral erosion; strong earthquake area; model experiment; erosion pattern





## 1. Introduction

After the "5.12 Wenchuan" earthquake in 2008, the mountains in the strong earthquake area were severely disturbed, adverse geological disasters occurred frequently, and the reserves of loose rock and soil mass generated by gully collapse and landslide increased sharply [1,2]. These deposits have poor particle sorting, a large pore ratio, and strong water permeability, which can easily initiate the formation of debris flows [3], resulting in an increase in the scale and frequency of debris flows [4]. This type of debris flow is mostly formed by the continuously increasing concentration of solids inside the fluid as the water in the channel erodes the channel bank. The experimental study of the lateral erosion of debris flow and the exploration of its disaster-causing mechanism can provide a better basis for the prediction and prevention of debris flow, so as to improve the level of debris flow disaster prevention and reduce the adverse impact of debris flow on gully erosion [5,6].

As hydraulic debris flow has a stronger erosive capacity to gullies, its occurrence ratio is increasing year by year [7]. The gullies are gradually deepened under the erosion of debris flow [8], and their boundary conditions change. The gully bank deposits are more likely to lose stability and fall into the gullies under the lateral erosion of water flow, which widens the gullies and sharply increases the scale of debris flow [9–11]. Through the analysis of debris flow data in Kansia Basin, Simoni et al. found that when the longitudinal slope of

the river channel exceeds 16 degrees, the river channel will produce lateral erosion [12]. G. J. Hanson et al. studied the erosion resistance of different parts of the riverbed through an on-site spraying experiment [13]. Zhou [14] and others found that riverbed erosion, bank collapse, and river widening caused by erosion are the main reasons for the triggering and scaling up of debris flow in the lower reaches of Wenjia Valley. The 2017 Wenchuan Yangtang ditch debris flow showed a strong lateral erosion widening effect on the ditch bank during the flow through the circulation area, the widening width generally reached 8~10 m, and the amount of material source squared taken away by lateral erosion along the course was as high as $3.0 \times 10^5$ m$^3$ [15]. During the debris flow movement in the 2019 Xiazhuang Valley ditch, the foot of the terrace slope on both banks of the ditch channel gradually panned back under the lateral erosion effect of the mudslide, which made the soil deposited on the shore at an early stage collapse in block form to converge in the debris flow [16]. It can be seen that material generated by lateral erosion of gully banks plays an increasingly important role in the formation of post-earthquake debris flows [17–19]. Through the relevant literature, we know that there are many factors affecting gully erosion, such as geology, landform, elevation, slope, soil characteristics, rainfall, etc. [6,20,21], which will have a great impact on gully erosion. In addition, Paramita Roy et al. also assessed the importance of gully erosion according to the influencing factors of gully erosion and drew the gully erosion sensitivity map of the Hinglo River Basin of eastern India, which can help the land management department to control the potential erosion area in advance, and also make full use of land resources to promote the sustainable development of the basin [22]. Pan Huali et al. [15] analyzed the erosion pattern and factors influencing the movement of debris flow in the channel and found that lateral erosion is particularly strong at the concave bank of the channel, and the foot of the channel bank is continuously retreating under the continuous erosion of debris flow, which makes the channel bank form a suspended body and collapse under the action of gravity, thus widening the channel laterally. Zhu Xinghua et al. [23] gave calculation formulas for lateral erosion occurrence, occurrence type, and transport rate through theoretical analysis and field in situ experiments, and classified lateral bank erosion types into dumping, falling, and slip collapse, and performed mechanical equilibrium analysis for each mode by analyzing the form of lateral bank fractures. Qu YP et al. [24] investigated the channel initiation mechanism and movement characteristics of hydraulic-type debris flows under different combinations of flow velocity, flow capacity, and longitudinal slope gradient in a hydraulic-type debris flow wash-out scale model experiment, but the influence of debris flow density on the results was not considered during the experiment. Zhao Yanbo et al. [25] explored the rule of erosion depth of debris flow from the gully bed under the conditions of gully slope, debris flow unit weight, and gully bed deposit gradation through a flume test. Chen Jing et al. [9] considered the erosion rate and depth of debris flow with different unit weights under different accumulation water contents and different slopes in a flume experiment of influencing factors of debris flow bottom erosion; however, neither Zhao Yanbo et al. [25] nor Chen Jing et al. [9] conducted in-depth research on the erosion width and volume of the accumulation body.

To date, many researchers have studied the erosion mechanism of debris flow through experiments, theories, and numerical simulation. However, their research objects are mainly focused on undercutting erosion. Therefore, the lateral erosion of gullies needs to be studied further [20,26,27]. Aiming at the problems of insufficient research on the mechanism of lateral erosion of debris flow and incomplete consideration of relevant influencing factors, this paper, on the basis of previous experimental research, carries out 18 groups of incomplete orthogonal experiments to explore the influence of the volume weight of debris flow, the slope of channel, and the gradation of deposits on lateral erosion and carries out analysis of the data on the shape, width, and square of deposits after erosion. The influence of different factors on the lateral erosion degree of debris flow is studied. The results are of great significance for further theoretical research on the side erosion of

debris flow, prediction of debris flow disasters, and improvement of the recognition and prevention of debris flow in strong earthquake areas.

## 2. Experiment Scheme Design

### 2.1. Experimental Apparatus and Equipment

The independently designed debris flow bed flume model experimental device [28] that can change the slope from 0 to 45° is used for the experiment. The experimental model is composed of four parts: a lifting device, mixing device, flume device, and stacking platform, as shown in Figure 1. The left side of the model device is a 6 m long and 0.3 m wide water tank, to stabilize the flow pattern; the first 4.2 m is set as the fixed bed section and the last 1.8 m is set as the dynamic bed section.

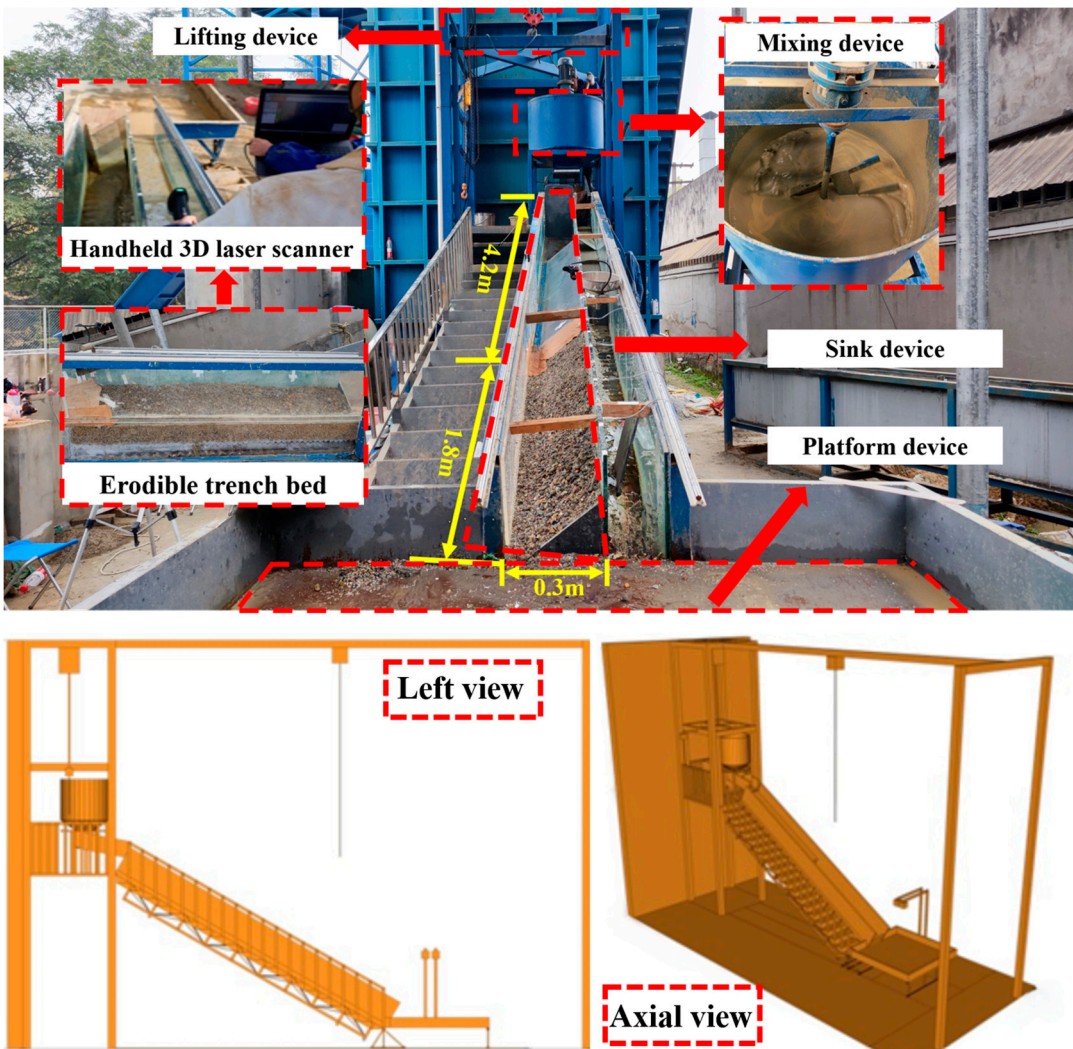

**Figure 1.** Experimental device.

### 2.2. Experimental Parameters

This experiment belongs to the generalized flume model experiment. The unit weight of debris flow in the strong earthquake area ranges from 1.6 g/cm$^3$ to 2.0 g/cm$^3$, the longitudinal slope of the gully varies from 7.6° to 42.8°, and the average longitudinal slope is 21.8° [24,29]. In order to reflect the flow characteristics of debris flow in the field, the volume weight range of debris flow configured in the experiment is consistent with field data. At the same time, considering the water infiltration under different longitudinal slopes of the gully in order to minimize the impact of water infiltration on the unit weight

of debris flow before erosion of the gully; referring to the relevant literature, the variation range of the longitudinal slope of the experimental gully is 8° to 20° [30].

Three kinds of debris flow with different unit weights, namely, rarefaction (1.6 g/cm$^3$), transition (1.7 g/cm$^3$), and viscosity (1.8 g/cm$^3$), are set up in the experiment. The materials for preparing the debris flow are natural river sand and clean water, and bentonite is added to improve the viscosity and workability of the debris flow, to prevent the debris flow from stratification. The proportions of debris flow with different unit weights are shown in Table 1. To consider the erosion effect of debris flow on different graded accumulations, three types of accumulations with 25% water content and different contents of fine particles are selected in the experiment. The content of fine particles in the accumulations at different levels is shown in Table 2.

**Table 1.** Composition of debris flow.

| Density | Water | Soil | Sand |
|---|---|---|---|
| 1.6 g/cm$^3$ | 0.35 | 0.1 | 0.55 |
| 1.7 g/cm$^3$ | 0.29 | 0.08 | 0.63 |
| 1.8 g/cm$^3$ | 0.24 | 0.06 | 0.7 |

**Table 2.** Fine particle content of accumulation.

| Gradation Types | Grading I | Grading II | Grading III |
|---|---|---|---|
| Content of fine particles (<1 mm) | 11.40% | 0.00% | 28.82% |

*2.3. Experimental Scheme*

In the experiment, the unit weight of debris flow, the longitudinal slope of the gully, and the gradation of accumulation are used as variables to study the influence of each variable on the lateral erosion of debris flow. Due to a large number of experimental variables, the experimental scheme is set by the method of incomplete orthogonality. Given the widespread existence of narrow and steep gully-type debris flow in the strong earthquake area, this type of gully is severely eroded and has a strong disaster-causing capacity. Its outbreak frequency increases significantly after an earthquake, and the slope of the gully is concentrated at about 16°. Therefore, the experiment takes 16° as the main slope to study the erosion law of different accumulation gradations under this slope. In the pre-experiment process, it was found that the erosion capacity of the debris flow with a unit weight of 1.8 g/cm$^3$ was weak, which was mainly represented by siltation. Therefore, the erosion process was only studied under the condition of the 20° longitudinal slope. The test scheme is shown in Table 3.

**Table 3.** Experimental scheme.

| Number | Longitudinal Slope Gradient (°) | Fluid Density (g/cm$^3$) | Gradation of Accumulation |
|---|---|---|---|
| 1 | 20 | Water | Grading I |
| 2 | 16 | Water | Grading I |
| 3 | 12 | Water | Grading I |
| 4 | 8 | Water | Grading I |
| 5 | 20 | 1.7 | Grading I |
| 6 | 20 | 1.8 | Grading I |
| 7 | 20 | 1.6 | Grading I |
| 8 | 16 | 1.6 | Grading I |
| 9 | 12 | 1.6 | Grading I |
| 10 | 8 | 1.6 | Grading I |

**Table 3.** *Cont.*

| Number | Longitudinal Slope Gradient (°) | Fluid Density (g/cm$^3$) | Gradation of Accumulation |
|---|---|---|---|
| 11 | 16 | 1.7 | Grading I |
| 12 | 12 | 1.7 | Grading I |
| 13 | 16 | 1.7 | Grading II |
| 14 | 16 | 1.6 | Grading II |
| 15 | 16 | Water | Grading II |
| 16 | 16 | 1.6 | Grading III |
| 17 | 16 | 1.7 | Grading III |
| 18 | 16 | Water | Grading III |

## 3. Experimental Study on Lateral Erosion of Erodible Gully Bed

In natural debris flow gullies, due to the erodibility of the gully bed, the lateral erosion of debris flow will be affected by the undercutting of the gully. Exploring lateral erosion regularity of erodible gully beds is beneficial to further deepen our understanding of lateral erosion of debris flow in strong earthquake areas, eliminate potential hazards in advance, and reduce potential natural disasters in the future, which is of great significance to reduce the safety risks of gully basins [31,32].

### 3.1. Erosion Form Analysis

After scouring, the hand-held 3D laser scanner is used to obtain the point cloud data of the final erosion form, and Geo magic software is used to process the measured 3D point cloud data, establish a scouring form model, and obtain the section curve information at the maximum erosion width and depth.

(1)  Erosion degree analysis of debris flow unit weight change

The morphology and 3D comparison of the gully channel after erosion of grade I accumulation by the different unit weight of debris flow under 20° longitudinal slope conditions are shown in Figure 2, and the characteristic cross-section of the gully channel after erosion is shown in Figure 3.

The results show that the maximum erosion width of 1.6 g/cm$^3$ debris flow is 111.6 mm, the maximum erosion depth is 96.1 mm, and the erosion volume is 22,338.27 cm$^3$. The maximum erosion width of 1.7 g/cm$^3$ debris flow is 90.70 mm, the maximum erosion depth is 104.7 mm, and the erosion volume is 9263.56 cm$^3$. The 1.8 g/cm$^3$ debris flow has no lateral erosion, and the erosion depth of 10.4 mm is generated at the front end of the gully, and 39.8 mm thick siltation is generated at the rear end of the gully, with the siltation volume of 2057.83 cm$^3$. The maximum erosion width of clear water is 192.2 mm, the maximum erosion depth is 91.3 mm, and the erosion volume is 28,556.99 cm$^3$. As can be seen from the erosion pattern under the action of 1.6 g/cm$^3$ debris flow, the gully presents a cut and pull trough at the bottom of the whole section, and the bank slope at the pull trough is steepened by lateral erosion. The 1.7 g/cm$^3$ debris flow forms a punch hole at the front end of the gully, and the erosion at the rear end is weakened; 1.8 g/cm$^3$ debris flow shows siltation in the process. In this process, water eroded the entire side bank slope, and a large number of side bank material sources accumulated in the ditch, reducing the original side bank slope, widening the bottom of the ditch, and forming a maximum thickness of 88.9 mm at the position after x = 1300 mm. The erosion degree of different fluids under the condition of an erodible gully bed is in the order of strong to weak: water, 1.6 g/cm$^3$ debris flow, 1.7 g/cm$^3$ debris flow, 1.8 g/cm$^3$ debris flow.

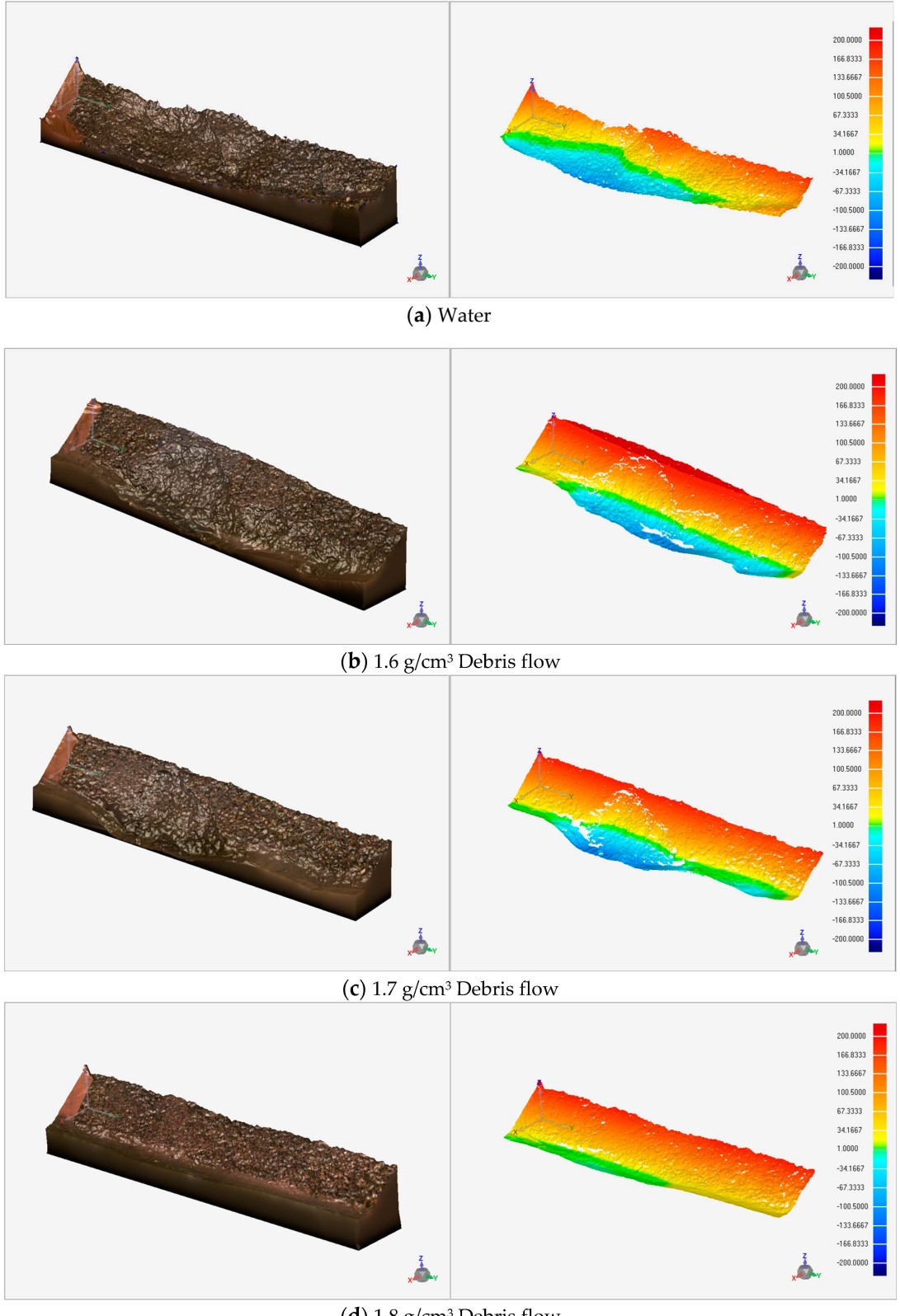

(**a**) Water

(**b**) 1.6 g/cm³ Debris flow

(**c**) 1.7 g/cm³ Debris flow

(**d**) 1.8 g/cm³ Debris flow

**Figure 2.** Comparison diagram of debris flow erosion with different unit weights.

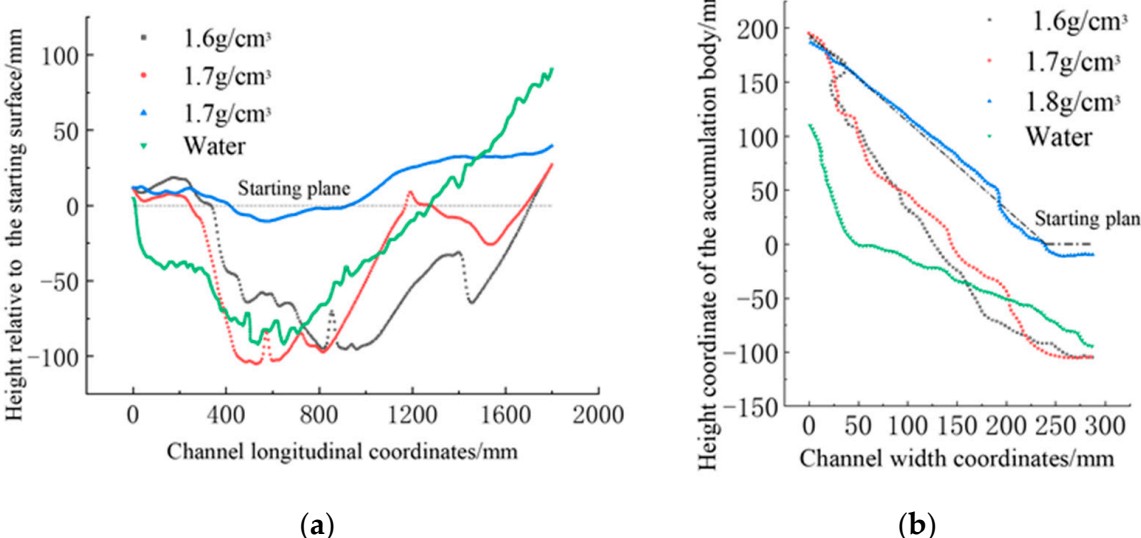

**Figure 3.** Characteristic cross-section after erosion of debris flow with different capacity. (**a**) Cross-section at maximum erosion depth. (**b**) Cross-section at maximum erosion width.

(2)    Analysis on erosion degree of gully longitudinal slope change

The morphology and 3D comparison of the gully channel after erosion of the graded I accumulation by different longitudinal slope conditions with a unit weight of 1.6 g/cm$^3$ debris flow are shown in Figure 4, and the characteristic cross-section of the gully channel after erosion is shown in Figure 5.

The analysis of the erosion of the lateral gully bank accumulation at different slopes showed that when the slope is 8°, the debris flow only forms scour holes at x = 300 mm, the maximum erosion width is 68.2 mm, the maximum erosion depth is 57.6 mm, and the erosion volume is 2244.01 cm$^3$; the rest of the locations are largely free of erosion. When the slope rises to 12°, the erosion degree gradually increases. At this time, the debris flow forms scour pits at x = 300 mm and x = 1200 mm, the maximum erosion width is 86.6 mm, the maximum erosion depth is 71.4 mm, and the erosion volume is 6347.80 cm$^3$. When the longitudinal slope of the gully is 16°, the debris flow forms a scour hole at the accumulation body x = 400 mm, and then a backward-pulling slot to form erosion on the entire section, which eventually leads to the steep side slope, with the maximum erosion width of 108.2 mm, the maximum erosion depth of 110.0 mm, and the erosion volume of 16,283.31 cm$^3$. When the gradient increases to 20°, under the action of debris flow, the gully is characterized by a full-section lower cut groove, the maximum erosion width is 111.6 mm, and the amount of erosion is 22,338.27 cm$^3$. Due to the instability of the bank slope at the position of the lower cut groove, particles flow into the eroded gully, resulting in the final erosion depth of 96.1 mm. In the actual erosion process, the erosion depth of debris flow under this unit weight reached 110.0 mm at the gully bed boundary. From the erosion situation, under the condition of the erodible gully bed, the erosion degree gradually increases with the increase in slope.

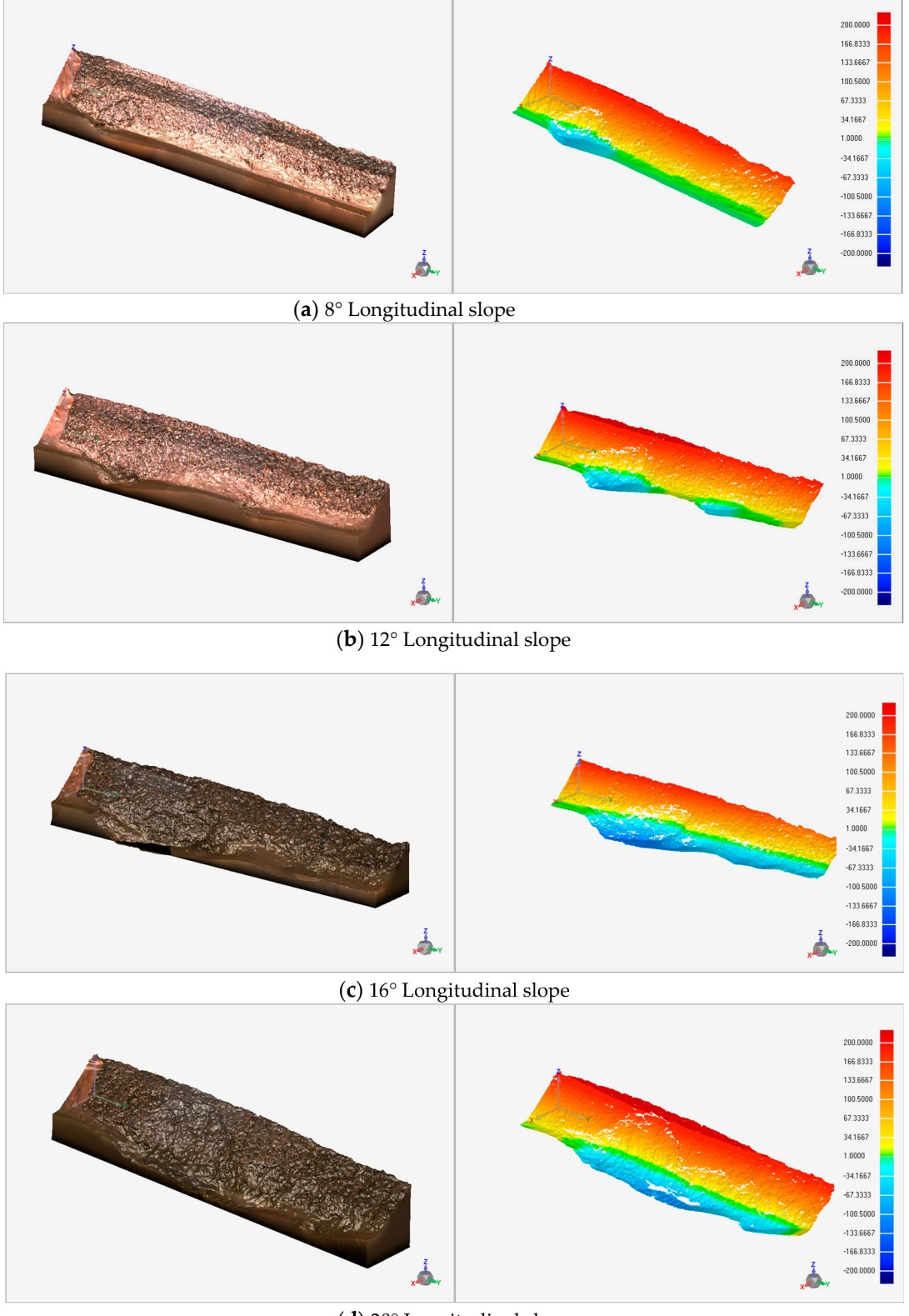

(**a**) 8° Longitudinal slope

(**b**) 12° Longitudinal slope

(**c**) 16° Longitudinal slope

(**d**) 20° Longitudinal slope

**Figure 4.** Comparison diagram of debris flow erosion with different longitudinal slopes.

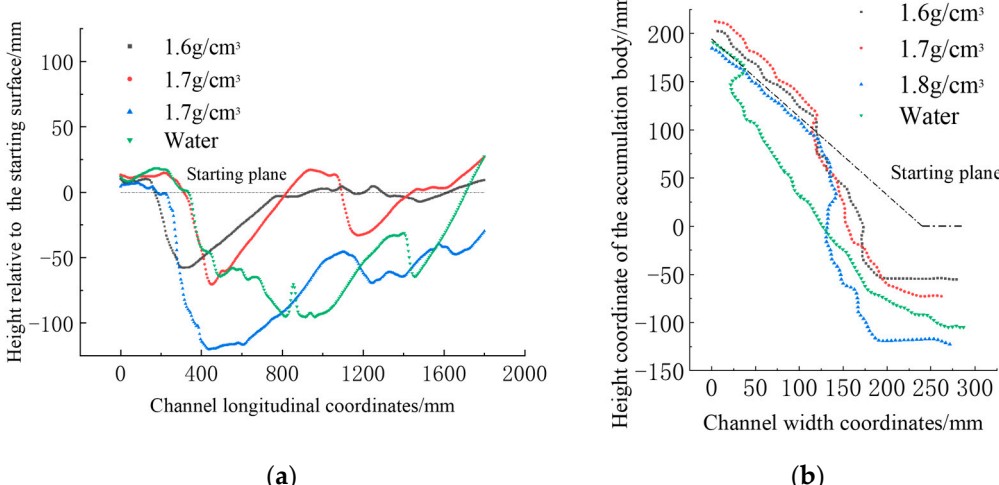

(**a**)                         (**b**)

**Figure 5.** Characteristic cross-sections after debris flow erosion under different longitudinal slope conditions (**a**) Cross-section at maximum erosion depth. (**b**) Cross-section at maximum erosion width.

(3)     Erosion degree analysis of different graded accumulations

The morphology and 3D comparison of the gully channel after erosion of different graded accumulation by the 1.6 g/cm$^3$ capacity debris flow under 16° longitudinal slope conditions are shown in Figure 6, and the characteristic cross-section of the gully channel after erosion is shown in Figure 7.

Under the condition of the erodible gully bed, the 1.6 g/cm$^3$ bulk weight debris flow produced a large degree of lateral erosion of the different graded accumulations, all accumulations reaching the boundary limit at the depth of erosion. In terms of erosion width of accumulations, the erosion extent of debris flow to graded I accumulations is the largest, with the erosion width reaching 108.2 mm, while the maximum erosion width of graded II and graded III accumulations are 83.5 mm and 48.3 mm, respectively. There is little difference in erosion volume among accumulations, with the erosion volume being 16,283.31 cm$^3$, 20,172.77 cm$^3$, and 18,932.91 cm$^3$, in turn. In terms of erosion pattern, the erosion pattern of debris flow to the three graded deposits is that scouring pits are formed near the channel longitudinal x = 450 mm, then scouring pits backward-pull grooves to form erosion across the whole section, which finally leads to the steepening of the entire side bank slope. The degree of erosion of the different graded accumulations shows that the fine grain content has a greater effect on the width of erosion and a smaller effect on the volume of erosion.

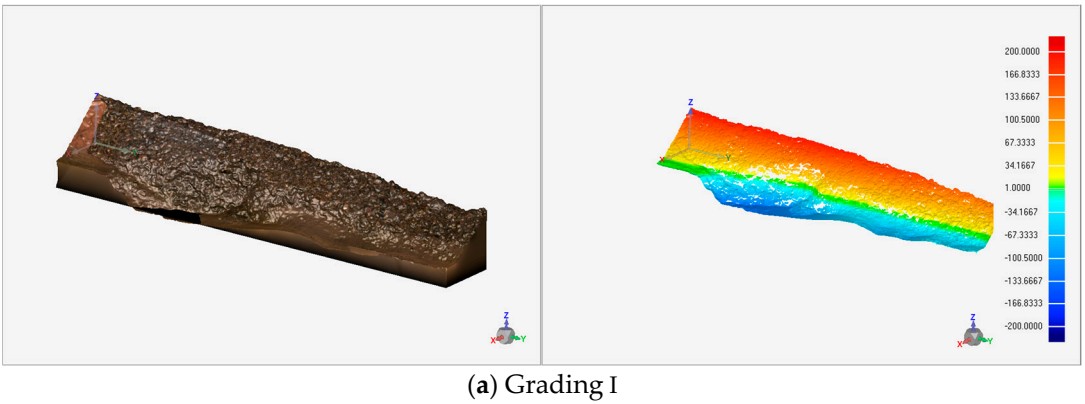

(**a**) Grading I

**Figure 6.** *Cont.*

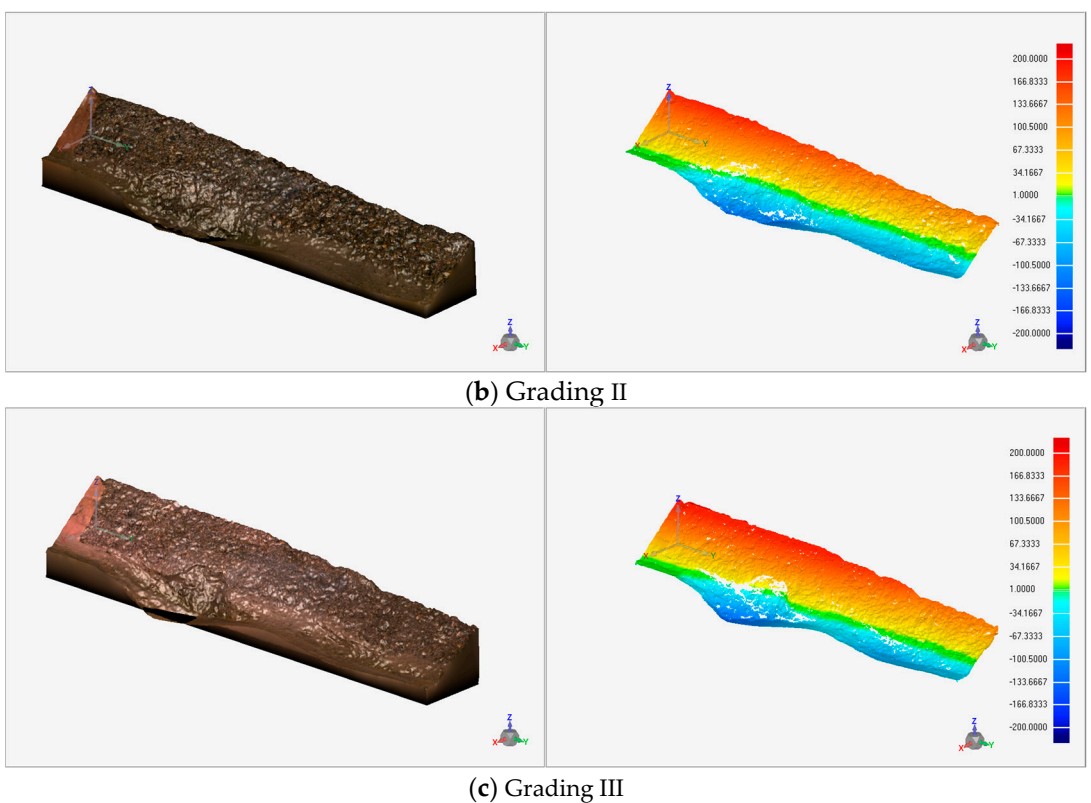

(**b**) Grading II

(**c**) Grading III

**Figure 6.** Comparison diagram of erosion of accumulation bodies with different gradations.

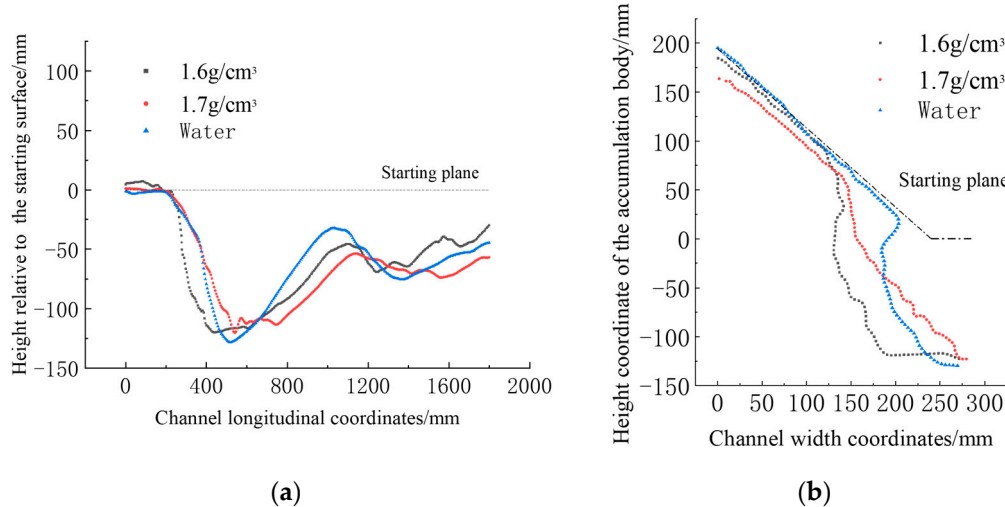

(**a**)　　　　　　　　　　　　　　　　　　　　　(**b**)

**Figure 7.** Characteristic cross-sections of different graded mounds after erosion by debris flow (**a**) Cross-section at maximum erosion depth.(**b**) Cross-section at maximum erosion width.

### 3.2. Analysis of the Width, Depth, and the Volume of Erosion

(1)　Erosion width

Under the condition of the erodible gully bed, the erosion width under different slope gradients is shown in Figure 8a. The erosion width of water and the 1.6 g/cm$^3$ debris flow are positively correlated with the channel slope. The 1.7 g/cm$^3$ debris flow does not show lateral erosion from 12° to 16°, but lateral erosion occurs when the slope is increased to 20°, suggesting a critical lateral erosion slope in the range of 16° to 20° for this volume of debris flow. The erosion widths of fluids with different bulk densities are in the following

order: water, 1.6 g/cm$^3$ debris flow, 1.7 g/cm$^3$ debris flow, indirectly indicating that fluid viscosity has a greater influence on the erosion width.

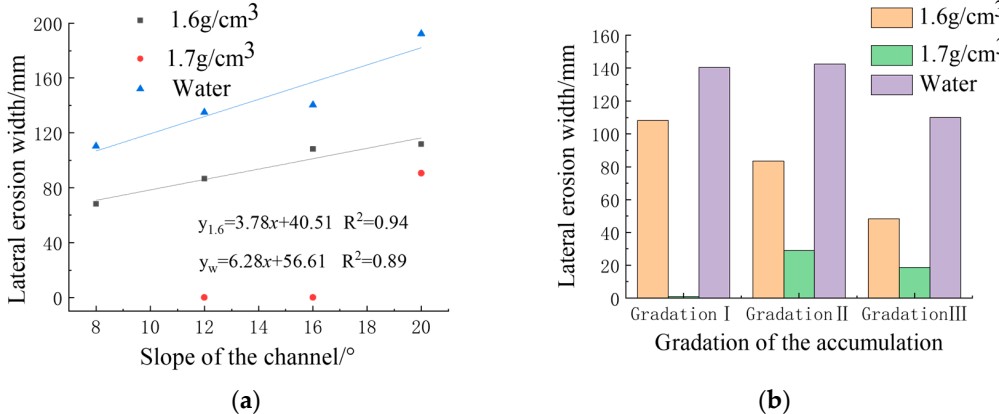

**Figure 8.** Lateral erosion width. (**a**) Erosion width under different slope gradients. (**b**) Erosion width under different graded accumulations.

The erosion widths of the different graded accumulations under 16° longitudinal slope conditions are shown in Figure 8b. Water and 1.6 g/cm$^3$ debris flow erode each channel greatly, while 1.7 g/cm$^3$ debris flow has a strong viscosity, low turbulence in the flow process, and low erosion width to the three graded deposits. The grade III accumulations have a higher content of fine particles than the other two grades, are more cohesive, and ultimately suffer the least erosion.

(2)    Erosion depth

The depth of erosion on different slopes is shown in Figure 9a. As the maximum erosion depth of the experimental design is 110 mm, the erosion depth reached the limit due to the influence of boundary conditions during the experiment; however, from the overall change degree, the erosion depth of each experimental group still increases with the increase in slope. However, under the condition of water and a 16° longitudinal slope, the final erosion depth is less than 12° longitudinal slope. This is because the channel accumulation collapses and slides at the end of the experimental process, causing the accumulation to block the bottom of the channel. The subsequent inflow is not enough to move the blockage body, making the final measured erosion depth smaller, but the actual erosion depth reaches the bottom of the channel.

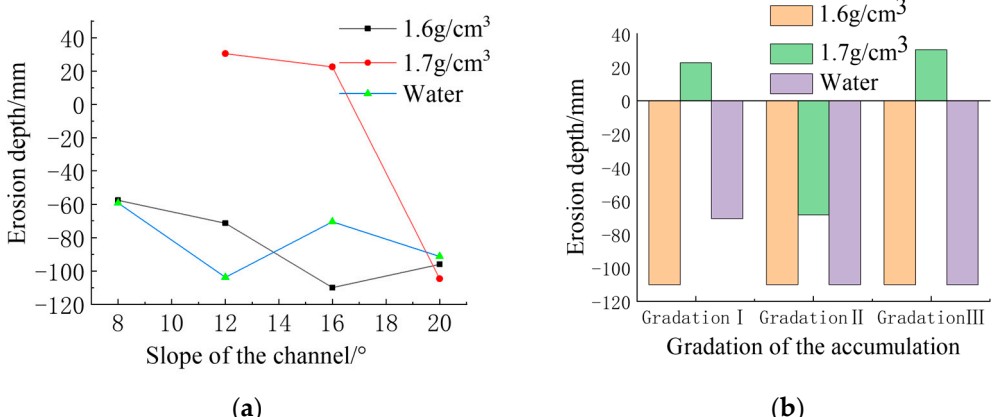

**Figure 9.** Lateral erosion depth. (**a**) Erosion depth under different slope gradients. (**b**) Erosion depth under different graded accumulations.

The depth of erosion of the different graded accumulations is shown in Figure 9b. Under the condition of the 16° longitudinal slope, the maximum erosion depth of water and the 1.6 g/cm³ debris flow reaches the boundary value. The 1.7 g/cm³ debris flow is silted under the conditions of grading I and grading III, and only under the conditions of grading II is incised, forming a 68.2 mm deep scour pit. The gradation II accumulation has a low content of fine particles, low cohesion, and large soil pores, which makes it easier for the debris flow to penetrate deeper into the mound, making the soil subject to undercutting erosion by floating forces. A correlation can be observed between the production of undercutting erosion and the fine particle content.

(3)    Volume of erosion

The total amount of erosion for each experimental group at different slopes is shown in Figure 10a. The clear water and 1.6 g/cm³ debris flow eroded under the condition of the 8° longitudinal slope, and the amount of erosion was positively correlated with the longitudinal slope gradient, with a high degree of correlation; for the 1.7 g/cm³ debris flow, under the conditions of 12° and 16° longitudinal slopes, only slight siltation occurs; under the conditions of 20° longitudinal slopes, erosion occurs in the gully, and the amount of erosion and siltation meets the exponential function relationship.

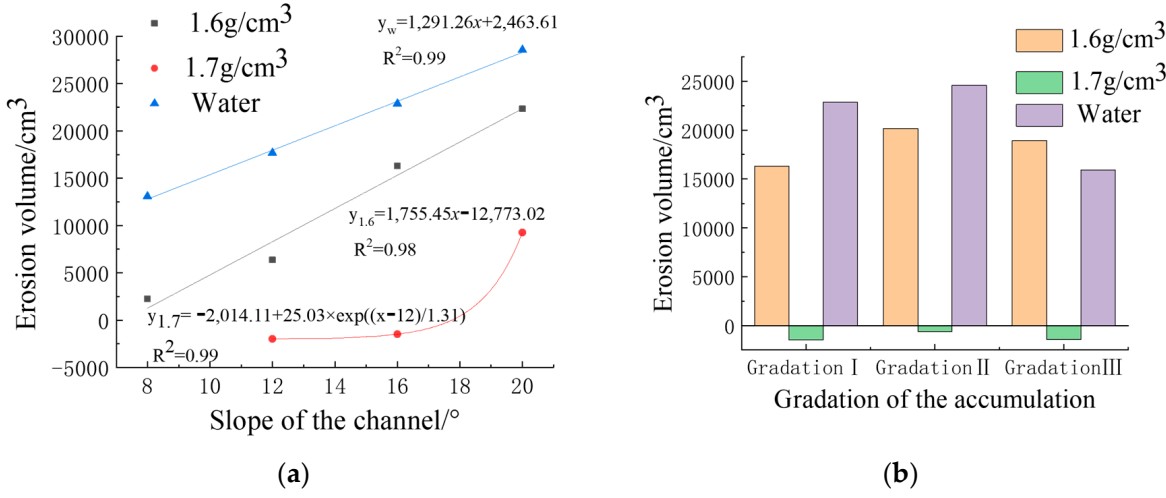

**Figure 10.** Lateral erosion volume. (**a**) Erosion volume under different slope gradients. (**b**) Erosion volume under different graded accumulations.

The total amount of erosion for different gradations is shown in Figure 10b. At the 16° longitudinal slope, 1.6 g/cm³ debris flow and clear water washed through the graded accumulations, while 1.7 g/cm³ debris flow silted up, and at this longitudinal slope, there was little difference in the total erosion of the different graded accumulations, and the gradation characteristics set in this experiment had little effect on the total erosion.

### 3.3. Multi-Factor ANOVA on Factors Influencing Side Erosion

When using SPSS software, the significance level is set as $p < 0.05$ and the larger the F value, the more significant the effect of the corresponding factor on the dependent variable; when the significance $p > 0.05$, the effect is not significant. With lateral erosion width and erosion volume as dependent variables and fluid density, slope, and accumulation grading as fixed factors, the Duncan model was selected to compare and analyze the main effects of each factor, and the results were as follows.

The impact analysis of each factor on the erosion volume and erosion width is shown in Table 4. Under the condition of the erodible gully bed, the volume of debris flow erosion and erosion width are affected by the same order of factors. The factor that had the greatest impact on the degree of erosion was fluid unit weight, followed by the gradient of the gully

longitudinal slope. It was further proved that under the condition of the erodible gully bed, the content of fine particles has no significant impact on the degree of gully erosion.

**Table 4.** Test for effect of each influencing factor.

| Factor | Erosion Volume | | Erosion Width | |
|---|---|---|---|---|
| | F | Significance *p* | F | Significance *p* |
| Density | 53.2988 | 0.000 | 35.450 | 0.000 |
| Slope | 7.994 | 0.007 | 9.224 | 0.004 |
| Graded | 1.922 | 0.202 | 2.883 | 0.108 |

## 4. Discussion

During the course of lateral erosion of channel-type debris flow in the strong earthquake area, the volume weight of debris flow, the longitudinal slope of the channel, and the gradation of deposits are three important factors affecting its lateral erosion. Taking these three factors as variables to carry out a tank model experiment is particularly important for studying the width, depth, and square of debris flow, and subsequent debris flow control and disaster prediction.

The volume weight of debris flow, the longitudinal slope of the channel, and the gradation of deposits have significant effects on the lateral erosion of debris flow. According to the model test and the results of data analysis, the factors influencing the width and volume of debris flow erosion are unit weight of debris flow, channel slope, and gradation of deposits, in turn. In addition, the gully slope provides power for debris flow, and the increase in the slope will also reduce the stability of the side bank slope, which will eventually lead to an increase in erosion volume and erosion width with the increase in the slope. In addition, the gully side bank is more prone to erosion damage. At the same time, it is observed in the test that the flow pattern of $1.6 \text{ g/cm}^3$ debris flow is relatively disordered, and it has a strong scouring ability to the side bank slope, while the $1.7 \text{ g/cm}^3$ debris flow is mainly laminar flow in the fixed bed section. After entering the side bank accumulation experimental area, the flow pattern evolves into a disturbed turbulent flow, and with the increase in slope, the turbulence degree intensifies, and the erosion to the gully gradually increases. At the same time, it was observed that the flow pattern of $1.6 \text{ g/cm}^3$ debris flow is relatively disordered, which has a strong scouring ability on the side bank slope, while the $1.7 \text{ g/cm}^3$ debris flow is mainly manifested as laminar flow in the fixed-bed section; the flow pattern evolved into the disturbed turbulent flow after entering the experimental area of side bank accumulation, the degree of turbulence increased with the increase in slope, and the erosion of the channel was gradually enhanced.

The results of erosion under erodible gully bed conditions show that when the slope is gentle, the debris flow will preferentially produce undercutting erosion of the gully bed during the flow through the gully, making the bank slope steeper, and thus inducing lateral erosion in the way of bank slope collapse and instability. The further intensification of erosion as the slope increases is due to the strong undercutting erosion, the expansion of the scale of the washout pit triggering a larger scale of lateral bank destabilization, and the more powerful transport capacity of the debris flow itself, which induces a large amount of channel material transport and eventually leads to the expansion of the erosion degree. The erodible ditch bed has a low degree of lateral erosion on the foot of the bank slope and a small erosion width, which makes it easier to produce a plugging effect when the loss of the lateral bank slope occurs, amplifying the scale of debris flow. Therefore, in the debris flow control project for the erodible gully bed, the prevention of undercutting erosion should be the main method, and on this basis, the protection of the bank slope foot should be strengthened.

This study, similar to other studies, has some limitations that cannot be ignored in the future [33], such as the constraints of the experimental site. The sink model used in this experiment is small and it is difficult to satisfy all similarity laws, so geometric similarity and boundary similarity were mainly considered during the experiment. In the future,

similar conditions such as debris flow movement and dynamics can be further considered by carrying out larger-scale model experiments. Due to the limitation of model size, the thickness of the experimentally set erodible layer is 110 mm, and the undercutting erosion reached the model boundary during the erosion process, but from the overall degree of change, the erosion depth of each experimental group still shows an increasing trend as the slope increases. In the future, we can deepen the thickness of the erodible layer in the gully bed and study the effect of undercutting erosion on the lateral erosion of the gully channel to further reveal the mechanism of debris flow erosion.

## 5. Conclusions

Based on the unit weight of debris flow, the longitudinal slope of the channel, and the gradation of deposits as variables, 18 groups of lateral erosion model experiments of debris flow were carried out. The effect of each variable on the lateral erosion of debris flow was studied by collecting data on the scouring process and the channel morphology of debris flow during the experiment. The main conclusions are as follows:

(1)  Both debris flow erosion volume and width increased with increasing gully slope, with a good linear fit correlation. This indicates that the gully side bank slopes are more susceptible to erosion damage under the increasing slope. The size of erosion volume and erosion width of the gully by different fluids are increased by clear water, 1.6 g/cm$^3$ mudflow, and 1.7 g/cm$^3$ mudflow, in order. In terms of the erosion volume and erosion width of the accumulation, the most severe erosion was observed for grade I. This indicates that a greater or lower number of fine particles are not conducive to the occurrence of lateral erosion in the gully, and this also provides direction for lateral erosion prevention and cure in the gully.

(2)  Compared with rigid riverbeds, lateral erosion of erodible riverbeds changes from slope foot scouring to undercutting erosion, which leads to instability of the riverbank. In addition, the lateral erosion pattern of the debris flow can be summarized as: the debris flow forms a wash pit through undercutting erosion, and mixes and shears in the wash pit to enable the side bank slope to form a critical surface, resulting in the side bank slope instability under the combined effect of debris flow infiltration, erosion, and gravity into the debris flow, and finally the scale of debris flow increases.

(3)  Based on the multi-factor ANOVA analysis of experimental data, the total erosion and erosion width were influenced by each factor in the following order of magnitude under erodible trench bed conditions: fluid capacity, trench longitudinal slope, and fine particle content of the accumulation.

As a natural disaster, debris flow is widely distributed in some areas of the world with special topography and geomorphological conditions. By studying erosion characteristics and investigating the influence of different factors on erosion, we can provide a better basis for prediction and prevention of debris flow. In the future, it is also possible to combine the degree of significance of each factor on debris flow erosion to map the erosion sensitivity of the watershed, such that potential risk areas can be treated in advance to reduce the adverse effects of debris flow erosion and promote the sustainable development of the watershed. Therefore, it is essential to explore the erosion characteristics of debris flow.

**Author Contributions:** J.Z. conceived the manuscript; J.Z. and D.L. dealt with point cloud data and drafted the manuscript; H.L., L.P. and Q.Y. provided funding support and ideas; J.Z. and D.L. conducted a field investigation and provided field data; Q.Y., H.L. and L.P. helped to improve the manuscript. All authors have read and agreed to the published version of the manuscript.

**Funding:** This research was funded by the Sichuan Provincial International Science and Technology Col-laboration and Innovation Project, grant number 2022YFH0078; the National Key R&D Program of China, grant number 2018YFC1505402; the National Natural Science Foundation of China, grant number No. 51809188" and "The APC was funded by Qiang Yao".

**Data Availability Statement:** The data used to support the findings of this study are available from the corresponding author upon request.

**Acknowledgments:** The authors would like to thank Qiang Yao, Hongtao Li, Liang Pei, and Dengze Luo for their suggestions on the manuscript and data analysis.

**Conflicts of Interest:** The authors declare no conflict of interest.

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
