# Peer review of "Experimental Study on Gully Erosion Characteristics of Mountain Torrent Debris Flow in a Strong Earthquake Area"

_water, doi:10.3390/w15020283_

Round 1
Reviewer 1 Report
Manuscript ID: water-2077801
The manuscript has some shortcomings which need to be improved prior to its publication. My recommendation is that the article needs Major Revisions before it can be considered for publication.
1. Abstract: The abstract is a bit generic. Please add some more information regarding your results.
2. Introduction is generalized. I would recommend following recent research articles to reconstruct with extensive literature review on prediction analysis and mention why you have chosen this topic for study.
“Implementation of Artificial Intelligence Based Ensemble Models for Gully Erosion Susceptibility Assessment
”
“Novel ensemble of multivariate adaptive regression spline with spatial logistic regression and boosted regression tree for gully erosion susceptibility”
”
“Gully erosion and climate induced chemical weathering for vulnerability assessment in sub-tropical environment”
“Evaluation of different DEMs for gully erosion susceptibility mapping using in-situ field measurement and validation”
3. Methodology section is weakly written. Literature on Use of RS-GIS in recent research is so little so my suggestion to reconstruct it with some recent articles which are published in 2015-2020. Author can take help from following articles:
“Hydrogeochemical characterization based water resources vulnerability assessment in India's first Ramsar site of Chilka lake”
“Hydrogeochemical Evaluation of Groundwater Aquifers and Associated Health Hazard Risk Mapping Using Ensemble Data Driven Model in a Water Scares Plateau Region of Eastern India”
“Characterization of groundwater potential zones in water-scarce hardrock regions using data driven model”
“Application of novel data-mining technique-based nitrate concentration susceptibility prediction approach for coastal aquifers in India”
4. In conclusion section, you have to mention the implications of your research and how it makes a footprint in scientific research. Try to incorporate your work to global interest how this research has worldwide importance. It will be interesting for the readers.
5. Reference: Re-check the whole reference just to make sure you have added all the references that you cited in your manuscript.
6. Improve the resolution of figure 1 that will help the readers for better understanding.
7. Apart from this the quality of the overall paper is very good. I prefer this article with acceptable with major modifications.

Author Response
Dear Editor and Reviewers,
Thanks very much for taking your time to review this manuscript. I really appreciate all your comments and suggestions! Please find my itemized responses below and the corresponding revisions/corrections in the re-submitted files.
Reviewer #1:
The manuscript has some shortcomings which need to be improved prior to its publication. My recommendation is that the article needs Major Revisions before it can be considered for publication.
- Abstract: The abstract is a bit generic. Please add some more information regarding your results.
Response: Thank you for your advice. We have deeply considered this problem and further improved the abstract.
Line 10-17: In recent years, as the frequency of debris flow outbreak in strong earthquake areas has increased and the scale has been expanding, in order to explore the erosion characteristics of debris flow, a lateral erosion flume model experimental device has been designed, and 18 groups of incomplete orthogonal experiments have been carried out, with the unit weight of debris flow of 1.6~2.0 g/cm3, the content of fine particles in the accumulation of 0~28.82%, and the longitudinal slope gradient of the gully of 8 °~20 ° as variables. The results show that the erosion width, erosion depth and erosion volume decrease with the increase of fluid bulk density and increase with the increase of gully slope.
Line 19-23: Finally, through the multi factor variance analysis, it is found that the order of the three factors on the gully lateral erosion degree from strong to weak is: debris flow unit weight, gully slope, and accumulation grading. The analysis results further show that the unit weight of debris flow has the greatest impact on the erosion degree of side slope, which is consistent with the experimental results.
- Introduction is generalized. I would recommend following recent research articles to reconstruct with extensive literature review on prediction analysis and mention why you have chosen this topic for study.
①:Implementation of Artificial Intelligence Based Ensemble Models for Gully Erosion Susceptibility Assessment
②:Novel ensemble of multivariate adaptive regression spline with spatial logistic regression and boosted regression tree for gully erosion susceptibility
③:Gully erosion and climate induced chemical weathering for vulnerability assessment in subtropical environment
④:Evaluation of different DEMs for gully erosion susceptibility mapping using in-situ field measurement and validation
Response: Thank you for your constructive and helpful suggestion. We have carefully read these research articles and cited all of them to reconstruct the introduction and the reasons for choosing this topic.
Line 36-40: Through the experimental study on the lateral erosion of debris flow and the exploration of its disaster causing mechanism, it can better provide a basis for the prediction and prevention of debris flow, so as to improve the level of debris flow disaster prevention and reduce the adverse impact of debris flow on gully erosion [5,6].
Line 61-68: Through the relevant literature, we know that there are many factors affecting gully ero-sion, such as geology, landform, elevation, slope, soil characteristics, rainfall, etc. [6, 20, 21], which will have a great impact on gully erosion. In addition, Paramita Roy et al. also assessed the importance of gully erosion according to the influencing factors of gully erosion, and drew the gully erosion sensitivity map of the Hinglo River Basin of eastern India, which can help the land management department to control the potential erosion area in advance, and also make full use of land resources to promote the sustainable de-velopment of the basin [22].
Line 89-92: At present, many researchers have studied the erosion mechanism of de-bris flow through experiments, theories and numerical simulation. However, their re-search objects are mainly focused on undercutting erosion. Therefore, the lateral erosion of gully needs to be further studied [20, 26, 27].
- Methodology section is weakly written. Literature on Use of RS-GIS in recent research is so little so my suggestion to reconstruct it with some recent articles which are published in 2015-2020. Author can take help from following articles:
(1):基于水文地球化学表征的印度首个奇尔卡湖拉姆萨尔湿地水资源脆弱性评估
(2):印度东部水惊吓高原地区地下水含水层水文地球化学评价及相关健康危害风险测绘
(3):基于数据驱动的模型表征缺水硬岩区地下水势带
(4):基于数据挖掘技术的硝酸盐浓度敏感性预测方法在印度沿海含水层中的应用
回应:感谢您的推荐。我们仔细阅读了这些论文,并引用了所有这些论文来改进手稿。
116-121行:为了反映野外泥石流的流动特性,实验中配置的泥石流体积重量范围与之一致。同时,考虑到沟壑不同纵向坡度下的水入渗情况,为尽量减少水入渗对沟壑侵蚀前泥石流单位重量的影响,参考相关文献,实验沟纵向坡度的变化范围为8°至20°[30]。
150-154行:探索可侵蚀沟床侧向侵蚀规律,有利于进一步加深对强震区泥石流横向侵蚀的认识,提前消除潜在危害,减少未来潜在的自然灾害,对降低沟壑盆地安全风险具有重要意义[31,32]。
374-377行:本研究与其他研究一样,也有一些未来不可忽视的局限性[33],如实验地点的约束,本实验使用的汇模型较小,难以满足所有相似性规律,因此在实验过程中主要考虑几何相似性和边界相似性。
- 在结论部分,您必须提及您的研究的含义以及它如何在科学研究中留下足迹。尝试将您的工作融入全球兴趣,该研究如何具有全球重要性。这对读者来说会很有趣。
回应:感谢您的建设性和有益的建议。我们在稿件中增加了相关内容。
411-419行:泥石流作为一种自然灾害,广泛分布于世界上一些具有特殊地形和地貌条件的地区。通过研究侵蚀特征,研究不同因子对侵蚀的影响,可以更好地为泥石流的预测和预防提供依据。未来还可以结合各因素对泥石流侵蚀的显著程度,绘制流域的侵蚀敏感性图,提前处理潜在风险区域,减少泥石流侵蚀的不利影响,促进流域的可持续发展。因此,探索泥石流的侵蚀特征至关重要。
- 参考文献:重新检查整个参考文献,以确保您添加了稿件中引用的所有参考文献。
回应:谢谢你的建议。我们已经检查了所有引文的格式。
- 提高图 1 的分辨率,这将有助于读者更好地理解。
回应:感谢您的建设性和有益的建议。 我们使用了最清晰的图片来修改图 1。同时增加了两个模型结构图,帮助读者理解。
- 除此之外,整体纸张的质量非常好。我更喜欢这篇文章,可以接受重大修改。
回应:谢谢你的鼓励。我们根据审稿人的意见对稿件进行了认真的修改,希望能被采纳。

Reviewer 2 Report
December 20, 2022
Dear Editor, dear Authors:
This manuscript entitled “Experimental Study on Gully Erosion Characteristics of Mountain Torrent Debris Flow in the Strong Earthquake Area” investigates the effects of multiple factors (as volume weight of debris flow, fine particle content and longitudinal slope of gully) on erosion of lateral blanks of gullies. The manuscript is well-written and capture the attention of the reader. I appreciate the structure and the formulation of text. Moreover, the topic of manuscript fits with the aim of journal and, in my opinion, it is ready for publication in an international journal as Water MDPI after some little improvements. Comments are reported in attached file.

Author Response
尊敬的编辑和审稿人:
非常感谢您抽出宝贵时间审阅此手稿。非常感谢您的所有意见和建议!请在下面找到我的逐项回复,并在重新提交的文件中找到相应的修订/更正。
审稿人#2:
这篇题为《强震区山洪泥石流沟壑侵蚀特征试验研究》的手稿研究了泥石流体积重量、细颗粒含量和沟壑纵向坡度等多种因素对沟壑侧向坯料侵蚀的影响。手稿写得很好,吸引了读者的注意力。我很欣赏案文的结构和措辞。此外,手稿的主题符合期刊的目的,在我看来,经过一些小的改进后,它已经准备好在国际期刊上发表,如Water MDPI。注释在附件中报告。
- 摘要:请重写句子。目前尚不清楚。
回应:谢谢你的建议。修订后的手稿作了以下修改。
10-17号线:近年来,随着强震区泥石流爆发频次增加,规模不断扩大,为探究泥石流侵蚀特性,设计了侧向侵蚀水槽模型实验装置,开展了18组不完全正交试验,泥石流单位重量为1.6~2.0 g/cm3、细颗粒物含量在积累中为0~28.82%,沟壑纵向坡度为8°~20°为变量。结果表明:侵蚀宽度、侵蚀深度和侵蚀体积随流体容重的增加而减小,随沟壑边坡的增加而增大;
回应:感谢您检查标点符号。 我们仔细地重新检查了手稿中的标点符号。
第 51 行:“;”已更改为“.”。
- 请使用上标
回应:感谢您指出这一重要问题。我们对稿件进行了修改和重新核对。
第 55 行:“3.0×105m3”已更改为“3.0×105 m3”。
回应:感谢您检查标点符号。
第 51 行:“;”已更改为“.”。
回应:感谢您检查标点符号。
第 76 行:“”已被删除。
- 研究员?科学家?请更改术语学者。
回应:感谢您的建议。 我们认为在稿件中用“研究者”代替“学者”更为合适。
第89行“学者”已改为“研究员”。
- 请调整空间和格式。
回应:感谢您指出此格式问题。 我们修改了标题“2.3 实验方案”的格式
- 指出此比例的含义,并使数字更清晰。
回应:感谢您指出这一重要问题。 该刻度由软件根据数值的变化默认,以便更好地区分每个区域的侵蚀。 我放大了图片中的数字,它们在单词文件中非常清晰。PDF文件中的图片可能会被压缩,这会降低稿件的清晰度。
- 请检查分号,它们通常处于不适当的位置。或者用在点上。
回应:感谢您检查标点符号。 我修改并进一步检查了全文中标点符号的使用。 纠正了分号的不当使用。
- 第176-178行:不清楚你的意思。使用逗号分隔大小写。
回应:感谢您指出这一重要问题。 我们对手稿进行了修改。
192-194行:可侵蚀沟床条件下不同流体的侵蚀程度按强到弱的顺序排列:水,1.6克/厘米3泥石流,1.7克/厘米3泥石流,1.8克/厘米3泥石流。
- 请检查格式。
回复:感谢您检查格式问题。 我已经统一了表达形式。
242行:图6“不同等级堆积体泥石流侵蚀对比图”改为“不同等级堆积体侵蚀对比图”。
- 也许您可以将方程式报告到表格中或使图表更具可读性。
回应:感谢您的建设性和有益的建议。 我们修改了图 10.a 中方程的位置,它比原始公式更具可读性。

Round 2
Reviewer 1 Report
This manuscript can be accepted in its present form